# From AI-Assisted *In Silico* Computational Design to Preclinical *In Vivo* Models: A Multi-Platform Approach to Small Molecule Anti-IBD Drug Discovery

**DOI:** 10.3390/ph18101536

**Published:** 2025-10-13

**Authors:** Joya Datta Ripa, Sarfaraz Ali, Matt Field, John Smithson, Phurpa Wangchuk

**Affiliations:** 1College of Science and Engineering, Cairns Campus, James Cook University, Cairns, QLD 4878, Australia; 2Centre for Tropical Bioinformatics and Molecular Biology, Cairns Campus, James Cook University, Cairns, QLD 4878, Australia; 3Immunogenomics Lab, Garvan Institute of Medical Research, Darlinghurst, NSW 2000, Australia; 4College of Medicine and Dentistry, Townsville Campus, James Cook University, Townsville, QLD 4811, Australia

**Keywords:** Inflammatory Bowel Disease, small molecules, *in vivo*, organoids, animal models, network pharmacology, bioinformatics, drug discovery

## Abstract

**Background**: Inflammatory Bowel Disease (IBD), including Ulcerative Colitis and Crohn’s Disease, is a multifactorial inflammatory condition of the intestinal tract driven by a complex interplay of genetic factors, immune system dysfunction, and gut microbiota alterations. This review aims to synthesize current advancements in modern drug development strategies for IBD. It emphasizes the integration of computational modelling, cell-based experiments, and animal model studies to enhance translational outcomes. **Methods**: To compile this review, an extensive literature search was performed utilizing PubMed, Scopus, and Google Scholar databases for English-language research and review articles published between 2000 and 2025 using keywords such as “IBD,” “molecular docking,” “bioinformatics,” “organoids,” “animal models,” and “network pharmacology,” among others. A total of 199 peer-reviewed studies were identified for inclusion based on relevance, transparency, and methodological robustness. **Results**: The review outlines a range of cutting-edge approaches to IBD drug discovery. These include computer modelling, molecular docking, and network analysis to accelerate early-stage target prediction and drug screening. The review further highlights the critical importance of utilizing 2D and 3D cell culture systems in parallel with advanced animal models. It emphasizes the critical integration of computational predictions with biologically relevant *in vitro* and *in vivo* validations to improve the reliability and efficiency of drug development. **Conclusions**: The integration of computer modelling, cell culture systems, and animal studies provides a revolutionary paradigm for accelerating drug discovery to IBD and other diseases enabling personalized and more effective treatment approaches.

## 1. Introduction

Inflammatory Bowel Disease (IBD) is a chronic, relapsing inflammatory condition of the gastrointestinal tract, with global rates significantly rising in recent years and associated healthcare costs of approximately USD 9000–12,000 per patient annually in high-income countries [1,2].

This intestinal inflammation is characterized by relapsing-remitting symptoms such as pain in the abdominal area, vomiting, chronic diarrhea, rectal bleeding, and gradual loss of weight [3]. These symptoms, often associated with fatigue and fever, may worsen the condition by turning into fistulas, infections, and even colorectal cancer [4]. It drastically impacts the life expectancy of the patients, leading to significant healthcare costs and overwhelming stress on the healthcare facilities [5]. IBD can be broadly categorized into Ulcerative Colitis (UC) and Crohn’s Disease (CD) based on the symptoms and pathogenesis [6]. UC is one of the most prevalent forms of IBD, characterized by repetitive lesions in the mucosal layer, primarily originating from the rectum and progressing through the colon [7]. Alternatively, CD is diagnosed with discontinuous features, submucosal lesions, patchy and transmural inflammation that can occur all over the bowel wall, resulting in fibrosis, strictures, and fistulas [8]. The pathogenesis of IBD involves host genetic predisposition, gut microbial dysbiosis, environmental influences, and immunological inconsistencies, although the precise etiology of IBD remains unidentified [9].

Despite significant efforts, therapeutic management of IBD remains challenging. Traditional therapeutic options for IBD including immunosuppressants, novel biologics such as anti-cytokine and anti-interleukin agents, and small-molecule inhibitors like Janus kinase (JAK) inhibitors exhibit limited long-term efficacy. Consequently, identifying novel drugs with improved therapeutic efficacy and higher safety profiles is imperative [10]. Currently, identifying target-based small molecules with good oral bioavailability and cell membrane permeability is considered an effective therapeutic option for tackling IBD, although the path to discovering such compounds remains a complex process [11]. To accelerate this process, next-generation IBD treatments need to increasingly incorporate advanced techniques, such as molecular modelling, computational studies, and bioinformatics, which streamline the discovery and development of novel small molecules [12].

Modern drug development utilizes integrative, technology-driven methodologies that merge computational (*in silico*) methods with experimental (*in vitro*) and preclinical (*in vivo*) models, building on recent advancements [13]. Among these, *in silico* methods have received significant acceptance for their capacity to diminish both time and expense in the preliminary phases of drug research. Molecular docking, pharmacophore modelling, virtual screening, quantitative structure–activity relationship (QSAR) analysis, network pharmacology, and machine learning are extensively used to predict drug–target interactions, evaluate pharmacological properties, and prioritize small-molecule candidates for further experimental validation [14]. Computational models offer flexibility and predictive capabilities; however, their efficacy depends on effective integration with experimental systems [15]. *In vitro* techniques, such as 2D and 3D cell cultures, patient-derived organoids, and gut-on-a-chip systems, provide physiologically pertinent information regarding human-specific drug responses, permeability, toxicity, and immunological effects [16]. Similarly, *in vivo* models, especially murine colitis models, provide essential insights into pharmacodynamics, systemic responses, and therapeutic efficacy within intricate biological systems [17].

Despite significant advancements, several prominent research gaps exist. Notably, the understanding of the interconnections between host-microbiome interactions and molecular signaling pathways in IBD remains insufficient. This hinders the development of targeted personalized medicines [18]. Furthermore, whereas cell culture and animal models are extensively employed for preclinical evaluation, many of these models lack physiological relevance, resulting in translational failure [19].

Moreover, *in silico* methodologies are frequently underexploited or isolated from experimental workflows, and there exists a deficiency of integrated frameworks that consolidate computational and lab-based strategies in IBD drug discovery [20]. To overcome these problems, there is a growing need for integrated drug discovery frameworks that combine bioinformatics, computational chemistry, and experimental biology into a single drug discovery pipeline [21]. Such methods enable iterative refinement, where computational predictions are tested in the lab, and the results of these tests are used to improve computer models. This creates a feedback loop that makes the models more accurate, reproducible, and useful for real-world applications [22]. Recent progress in systems pharmacology, AI-assisted screening, and omics-based databases has enhanced the capacity to discover innovative treatment options specifically suited to the intricate pathophysiology of IBD [23]. This review aims to elucidate the strategic integration of modern computational and experimental models to expedite the discovery of small-molecule drugs in IBD.

Overall, it provides an in-depth insight into how modern methodologies spanning computational, *in vitro*, and *in vivo* domains are being harnessed and integrated in the field of IBD drug discovery. It aims to foster more collaborative and translational approaches that link computational understanding to experimental and clinical significance by highlighting present strategies, constraints, and future potential. Its purpose is to direct future research and promote translational approaches that connect fundamental biology to clinical application.

## 2. Review Methodology

This narrative review examines the development and implementation of modern drug discovery methodology in IBD, highlighting the tools and frameworks that are influencing next-generation therapies. It analyzes essential *in silico* methods employed for target prediction, ligand screening, and compound optimization, in conjunction with sophisticated *in vitro* platforms, including organoids and microfluidic devices, that more precisely replicate the intestinal microenvironment. The critical significance of *in vivo* animal models in preclinical evaluation is also discussed, along with *in vitro* methods that utilize human cells and sophisticated tissue cultures. Finally, the review examines the uses, advantages, and disadvantages of each strategy for identifying new targets and evaluating potential drug candidates, aiming to provide a consolidated *in vitro*, *in vivo*, and *in silico* approach to the evolving field of IBD drug discovery and its future directions.

For this narrative review, we conducted a structured, comprehensive literature search of research articles and reviews published between 2000 and 2025. We utilized specific key words and search terms including “Inflammatory Bowel Disease”, “IBD”, “Small molecules”, “Molecular docking”, “Bioinformatics’, “Network pharmacology”, “Computational modelling”, “Mathematical modelling”, “Software database”, “*in vitro* model”, “Organ-on-a-chip”, “Animal model”, “*In vivo* IBD animal model”, “Combined *in vitro in vivo in silico* model IBD.” Studies obtained from PubMed, Google Scholar, and Scopus were thoroughly evaluated according to established inclusion and exclusion criteria. Only English-language, accessible full-text articles relevant to IBD drug development were retained, while duplicates and irrelevant research were excluded (Figure 1).

The above search terms were combined using the Boolean operators “OR” and “AND”. The preliminary keyword search on Google Scholar, PubMed, and Scopus for the years 2000–2025 yielded 149,000, 111,343, and 104,450 records, respectively. After eliminating duplicate records in the databases, there were about 5000 unique records left. After excluding non-English publications and inaccessible full-text articles, the figure decreased to approximately 2000. Further screening to remove articles not directly relevant to IBD therapeutic development resulted in approximately 500 records. Finally, after evaluating methodological quality and excluding outdated research or those with inconsistent findings, 198 research articles and review papers were ultimately chosen for inclusion in this review.

## 3. Modern Drug Discovery Approaches in IBD Research

Modern drug development for IBD focuses on identifying new therapeutic targets and developing effective treatments by combining advanced cutting-edge technologies. These encompass biomarker-based stratification, virtual screening, molecular docking, bioinformatics, and network pharmacology resources that support conventional *in vitro* and *in vivo* models in IBD research. The integration of different tools is facilitating the development of more personalized and mechanistically grounded treatment approaches, as summarized in Figure 2 and subsequent sections.

### 3.1. Discovery of Novel Biomarkers in IBD

#### 3.1.1. Diagnostic, Predictive, and Monitoring Biomarkers

Biomarkers, such as the presence of a particular protein or a motif of gene expression, are quantifiable indicators of a biological condition used to identify patients risk of developing the disease, predict how well a treatment will work, or evaluate the activity of the illness [24]. Biomarkers can be obtained from various biological layers, including metabolomics, proteomics, transcriptomics, and microbiome profiling, each representing distinct aspects of disease activity and immunological dysfunction [25]. In IBD research, biomarker design has transformed how clinicians and researchers diagnose the appropriate type of IBD, track therapeutic response, and understand individual IBD conditions [26]. The biomarkers summarized in Table 1 were selected based on a literature search, focusing on current clinical and translational research, as well as potential future applications. For clarification, biomarkers are categorized into established markers already used in clinical practice (fecal calprotectin, CRP), and research-stage markers under early clinical validation (e.g., miRNA panels, REG3α, CPa9-HNE), and emerging experimental markers in preclinical investigation (RUNX1 and MPO). This categorization highlights both the current clinical toolkit and the frontiers of biomarker discovery in IBD.

These biomarkers can aid in the detection, prognosis, and monitoring of IBD development through various mechanisms. Current biomarkers, including metabolomic, proteomic, inflammatory, and microbiota-derived indicators, are being utilized in clinical or research settings to diagnose IBD, evaluate disease activity, monitor therapy efficacy, and forecast recurrence. Currently, diagnostic biomarkers such as Fecal calprotectin and CRP are commonly used as non-invasive indicators of intestinal inflammation. In contrast, miRNA panels are gaining recognition for their ability to differentiate between UC and CD, as well as to predict susceptibility to biologic therapy [25].

Predictive and prognostic biomarkers represent a growing area of investigation. This encompasses transcriptomic and serological markers, including RUNX1 and CPa9-HNE, which elucidate the molecular progression of the disease and its crucial role in the etiology of colorectal cancer. Although many of these biomarkers are still in preclinical or early clinical stages, they are anticipated to enhance disease classification and inform personalized treatment strategies. Collectively, these current and novel biomarker designs are transforming the domain of IBD care, shifting it from reactive therapy to proactive, individualized interventions [34,35,39].

Monitoring biomarkers is a significant tool for tracking disease progression and treatment response. Along with well-known biomarkers like fecal calprotectin and CRP, emerging candidates such as myeloperoxidase (MPO) and REG3α may provide more precise information on neutrophilic inflammation and mucosal repair. These investigational candidates have the potential to enhance disease monitoring after clinical validation [38]. Overall, this landscape of IBD biomarkers spans from well-established diagnostic tools to experimental markers that hold promise for refining disease classification, prognosis, and therapeutic monitoring. Therefore, the categorization of different types of biomarkers is essential for translating discovery into practice and for setting realistic expectations regarding their current applicability. 

#### 3.1.2. Digital Health Tools and Computational Approaches in Biomarker Discovery

Alongside traditional biomarker research, digital health technology and computational methods are being applied to facilitate IBD monitoring and treatment personalization [40]. Digital health tools, including biosensors and wearable electronics, are designed to enable the transition from one-time tests to continuous monitoring of inflammation [41]. These technologies facilitate earlier identification and more accurate classification of disease subtypes, while allowing the prediction of outbreaks and the ongoing evaluation of therapy responses [42]. For instance, the SWEATSENSER platform is a wearable biosensor that can detect IL-1β and CRP in eccrine sweat. These cytokines, which are increased during IBD flares, are essential signs of inflammation in the mucosa. The device employs electrochemical impedance spectroscopy and demonstrates significant concordance with ELISA-based methodologies. This breakthrough offers real-time, noninvasive disease activity tracking, allowing for early flare diagnosis and rapid medication adjustment, both of which are critical components of personalized IBD treatment [43].

Moreover, recent advancements in computational biology have revolutionized biomarker discovery by facilitating the incorporation of multi-omics datasets, including metagenomics, meta proteomics, and metabolomics, to characterize the gut microbiota and its involvement in IBD pathogenesis. Machine learning and deep learning techniques now elucidate intricate microbe–host interactions, uncovering diagnostic and prognostic biomarkers. These models can forecast individual reactions to microbiota-targeted medicines and facilitate the creation of customized treatment strategies grounded in microbial, genetic, and metabolic profiles [44].

Importantly, AI-driven platforms are significantly altering small molecule drug discovery by integrating biomarker design with phenotypic screening to uncover compounds that influence disease-relevant pathways. Yu et al. devised a high-throughput, image-based phenotypic screening methodology that integrates inflammatory biomarker analysis with AI-driven profiling, successfully discovering small compounds that can reverse fibrotic phenotypes in intestinal myofibroblasts. Integrative techniques demonstrate the convergence of AI, biomarker design, and small molecule discovery to deliver personalized treatment for IBD [45]. Although encouraging, these findings have not yet been clinically validated. A systematic review by Chen et al. further demonstrates that multi-omics biomarkers can accurately predict the efficacy of biologic and small-molecule therapies in IBD, but these predictions remain in the early research phase and require prospective clinical validation [46].

### 3.2. In Vivo Studies in Small-Molecule Drug Discovery

While discovering biomarkers is the basis for precision medicine and patient categorization in IBD, advancing from biomarker discovery to therapeutic development requires more computational techniques. The employment of *in silico* models, particularly in the development of small molecule therapies, represents a significant leap forward in modern drug discovery. Until now, biologics have dominated the field of IBD therapeutics, as developing small molecules for IBD is still a complex process due to the involvement of diverse molecular mechanisms. Moreover, discovering a small molecule and bringing it from the bench to the bedside through the traditional drug discovery process takes approximately 10 to 15 years and a $1 billion to $3 billion investment [47]. In the last few decades, due to the advancement of computer algorithms and hardware, *in silico* drug discovery has emerged as a cost-effective, time-saving strategy, benefiting from the integration of cheminformatics and bioinformatics approaches. These approaches are classified into two main categories: (1) lead identification by cheminformatics and computer-aided drug design (CADD), and (2) target identification via bioinformatics, omics studies, and network pharmacology [48].

#### 3.2.1. Lead Identification via Cheminformatics and CADD

One of the most established cheminformatics methods is Quantitative Structure-Activity Relationship (QSAR) modelling. QSAR constructs mathematical models that forecast biological activity based on chemical structure, even when the mechanisms of action are not entirely elucidated. In complex diseases such as IBD, which involve multiple molecular mediators, QSAR is crucial for identifying and optimizing lead compounds. Its capacity to eliminate statistical outliers via self-consistent modelling also reduces overfitting, rendering it an efficient pre-screening technique [48]. Statistical modelling techniques such as multilinear regression analysis, linear discriminant analysis, and machine learning (ML) algorithms are frequently utilized to enhance QSAR performance. For example, Cong et al. used ML-based recursive feature elimination (RFE) to predict TACE inhibitors, molecules relevant to both rheumatoid arthritis and Crohn’s disease. 

These models primarily provide quantitative correlations rather than elucidating molecular pathways, which is a fundamental drawback of most statistical and machine learning approaches [48,49].

#### 3.2.2. Mathematical Modelling in IBD

Mathematical models, particularly those utilizing Ordinary Differential Equations (ODEs) and Partial Differential Equations (PDEs), have played a significant role in elucidating immune cell movements in murine colitis models through the analysis of longitudinal single-cell RNA sequencing and flow cytometry data. These models enable researchers to monitor colonic immune cell populations longitudinally and associate them with therapy outcomes [50]. Certain models, such as those suggested by Con et al., incorporate clinical characteristics into decision trees to direct therapy in acute severe ulcerative colitis (ASUC), forecasting the probability of colectomy [51]. Other models, such as the three-compartment model developed by D’Ambrosio et al., mimic the interaction between host, microbiota, and medication by evaluating the impact of Vedolizumab on mucosal barrier integrity. Despite their predictive capabilities, these models are often limited by their dependence on high-quality data and reduced generalizability to external populations [52]. Ultimately, mathematical models can bridge the gap between experimental research and patient care by guiding scientific breakthroughs and informing a precision medicine-based healthcare system in IBD research [53].

#### 3.2.3. Molecular Docking: An Essential Molecular Modelling Tool

Molecular docking is a popular structure-based molecular modelling technique for estimating the binding orientation and affinity of a ligand to its protein targets. It has two main steps: ligand conformational sampling and scoring based on predicted binding energies. Docking simulations are most useful when the binding pocket of the target is known, which may frequently be determined from existing crystal structures or homologous protein complexes [54]. Docking is commonly utilized in IBD research to screen drug libraries against critical inflammatory mediators, including JAK kinases, NF-κB, and integrins. Table 2 outlines various popular academic and commercial molecular docking tools and commercial platforms that support docking, visualization, and molecular modelling.

Molecular docking has become an essential computational method in the initial phases of small-molecule drug discovery for IBD [20]. A diverse selection of docking platforms, ranging from open-source academic tools to commercially licensed software, is often utilized to predict ligand-protein interactions and evaluate binding affinities [54]. The AutoDock Vina is often employed as an academic tool, favored for its compatibility with high-throughput virtual screening and user-friendliness [55]. LeDock has exceptional docking precision in semi-flexible ligand binding, making it particularly advantageous for lead optimization in small-molecule design [61]. SwissDock, utilizing EADock DSS algorithm, provides a user-friendly online interface for expedited docking, albeit it is primarily designed for the initial screening of small, rigid ligands [57]. Recent studies have demonstrated the implementation of SwissDock in IBD research. For example, Bibi et al. utilized SwissDock to assess bioactive compounds from *Acacia honey* and *Nigella sativa* oil, which were further experimentally validated in a murine colitis model. This study highlights the significance of SwissDock in identifying new IBD drugs [73].

Conversely, commercial software such as Glide (Schrödinger), GOLD, and the Molecular Operating Environment (MOE) offer superior scoring algorithms, adaptable receptor docking, and integrated ADMET filtering capabilities, making them exceptionally well-suited for structure-based drug discovery in complex diseases such as IBD. Glide is distinguished for its accuracy in simulating small-molecule interactions and its strong capability in replicating crystallographic poses. GOLD excels at modelling adaptable protein-ligand systems, whereas MOE provides an extensive array of features for pharmacophore modelling, homology modelling, and molecular simulations [66,67,69].

Despite these benefits, there are certain limitations of molecular docking. Many academic docking tools assume only rigid or semi-rigid receptor conformations, potentially misrepresenting binding interactions, particularly for the flexible proteins that undergo conformational alterations or undergo allosteric modulation [74]. 

Tools such as MedusaDock seek to mitigate this problem by integrating ligand and protein flexibility [60]. Software such as PyMOL, UCSF Chimera, and Discovery Studio Visualizer enhances visualization and post-docking analysis, broadly utilized for the interpretation of binding postures and molecular interactions [54]. Ensemble-based docking, utilizing multiple receptor conformations and flexible-receptor docking tools, along with post-docking refinement via molecular dynamics (MD) simulations, can be used to ensure the docking accuracy [75]. That being said, the selection of docking software for small-molecule discovery in IBD can significantly affect both efficiency and predicted accuracy. In general, docking tools should be determined according to the purpose of the study, the character of the target-ligand system, and the level of required precision. For the preliminary ranking of candidate molecules, AutoDock Vina is widely used for virtual screening of varied chemical libraries due to its rapidity, consistency, and straightforward parameter reporting [76]. For flexible protein targets, including kinases, transcription factors, and membrane receptors involved in IBD pathophysiology, commercial software like GOLD can successfully interpret conformational changes that other rigid docking tools may overlook [77]. Similarly, another commercial tool, Glide is superior for producing high-resolution poses and precise scoring, particularly in the validation of high-priority hits from first screenings by AutoDock Vina [78].

Additionally, when both ligands and receptors are flexible, MedusaDock can offer a distinctive benefit by directly simulating side-chain rearrangements, enabling a more accurate representation of dynamic binding sites [79]. Using these technologies together could provide a synergistic workflow for finding small compounds in IBD. For example, Vina can be utilized for large-scale screening, Glide for precise scoring, GOLD for flexible receptor optimization, and MedusaDock for elucidating dynamic interactions, which makes candidate therapeutic prioritization more reliable and biologically relevant. The use of these tools in conjunction with advanced approaches, such as molecular dynamics simulations, ensemble docking, and machine learning-based scoring functions, is becoming increasingly recognized as a best practice for enhancing prediction accuracy and guiding rational drug design in IBD [80].

#### 3.2.4. Target Identification Through Bioinformatics Approach: Network Pharmacology and Systems-Biology

Combining cheminformatics and molecular modelling, bioinformatics-driven target discovery represents a revolutionary methodology in IBD research. Network pharmacology (NP) is a promising strategy that employs a systems-level approach to design and analyze biological interaction networks, aiming to identify therapeutic targets through node centrality, network topology, and functional interconnectivity [81]. This NP paradigm provides an integrated bioinformatics and physiological network that can portray the interconnection between drug-disease-hub genes and guide the innovation and application of novel drugs [82].

NP has been highly beneficial in overcoming the shortcomings of traditional drug discovery pipelines, which often experience low translational success and high attrition rates. A systems pharmacology study by Sadegh et al. illustrated the repurposing potential of non-gastrointestinal drugs, including Imatinib, Fostamatinib, and Ruxolitinib, for the treatment of IBD, utilizing the NeDRexDB platform and MuST algorithm, emphasizing how integrative network analysis can uncover new indications for established compounds [83]. In addition to drug repurposing, NP models characterize the complexities of IBD-related pathways and nodes. One such study revealed 43 key nodes and over 300 interconnections within the IBD disease network, highlighting the polygenic and multifactorial aspects of IBD etiology [84].

Furthermore, NP functions as a translational bridge between conventional herbal therapy and modern therapies. For instance, while compounds from Suxin-Huangqin-Fang (SXHGF), Huangqin decoction (HQD), and Jasminum elongatum (JE) have shown anti-inflammatory effects, network pharmacology has been instrumental in elucidating their mechanisms. SXHGF was identified as a target for lipopolysaccharide production and reactive oxygen species pathways [85]. HQD influenced NF-κB, IL-17, and Th17 differentiation signaling [86], and JE mitigated colitis through the IκB/p65/COX-2/arachidonic acid pathway [87]. NP can also identify bioactive compounds from natural products that may be therapeutically potent for IBD [82]. This NP procedure mainly focuses on the following steps: 1. Identifying bioactive compounds. 2. Mapping the disease targets, hub genes, or proteins by networking. 3. Constructing Drug-Target-Disease Networks by applying protein-protein network construction and Pathway Enrichment Analysis: Gene Ontology (GO) and Kyoto Encyclopedia of Genes and Genomes (KEGG) enrichment analyses. 4. Analyzing Key Pathways and Mechanisms, and 5. Validating the data of closely connected pathways by molecular docking, *in vitro* and *in vivo* studies [82,88,89]. Notwithstanding its advantages, NP faces difficulties in the field of IBD drug discovery pathway, owing to the intricacies of multi-pathway interactions and the limited overlap across omics-derived networks. Advanced integration approaches are crucial for cross-validating results from genome, transcriptomics, proteomics, and metabolomics datasets [84,90].

#### 3.2.5. Other Bioinformatics Approaches in IBD Research

In parallel to network pharmacology, other bioinformatics technologies and tools are transforming IBD research by facilitating high-throughput, multi-dimensional analyses. Multi-Omics techniques like metagenomics, meta-transcriptomics, and meta-proteomics have enhanced our knowledge of microbial dysbiosis and its functional consequences in the pathophysiology of IBD [91]. For example, the Integrative Human Microbiome Project (iHMP) has shown how longitudinal characterization of the microbiome and host transcriptome might identify microbial changes that predict disease exacerbations in individuals with Crohn’s disease [92].

High-throughput omics technologies (HTTs), such as single-cell RNA sequencing and mass spectrometry-based proteomics, provide unparalleled investigation of IBD biology, enhancing biomarker identification, patient classification, and treatment response forecasting [93,94]. Nonetheless, obstacles such as diverse data formats, the absence of standardized workflows, and inter-study heterogeneity in omics interpretations persist. Integrating these datasets and corroborating results across platforms is crucial for enhancing the translational efficacy of omics-based bioinformatics in precision IBD therapy [94]. A comprehensive metagenomic study by Khorsand et al., utilizing datasets from the Inflammatory Bowel Disease Multi-omics Database (IBDMDB), showed elevated levels of multiple pathogenic species in IBD, including *Klebsiella pneumoniae*, *Proteus mirabilis*, and *Citrobacter freundii*, and highlighted disease-specific microbial signatures. Notably, *E. coli* biosynthetic pathways, such as aerobactin siderophore production, lipopolysaccharide (LPS) production, and enterobacterial common antigen synthesis, were all considerably elevated in CD. This indicates a transition towards pro-inflammatory microbial activity. The menaquinol synthesis routes associated with *K. pneumoniae* strains in UC were significant. These findings highlight the diagnostic and therapeutic significance of microbiome-based bioinformatics in IBD [95].

Alongside microbiological research, proteomics has become a promising tool for examining the molecular basis of IBD. A study by Arash et al. highlighted the use of proteomic technologies, including mass spectrometry and two-dimensional gel electrophoresis, for identifying differentially expressed proteins in intestinal tissues, serum, and fecal samples of individuals with IBD [96]. 

These proteins were linked to immunological modulation, oxidative stress, and epithelial barrier integrity, presenting potential indicators for diagnosis and treatment assessment. The study highlighted the potential of combining proteomic data with other omics platforms to improve precision medicine and personalized treatment in IBD, despite persistent challenges such as sample variability and the lack of standard operating procedures [96]. Notwithstanding the potential of multi-omics data, numerous obstacles persist. Varied data formats, absence of defined workflows, and inter-study variability in annotation complicate repeatability and integration. Multi-Omics Factor Analysis (MOFA), iClusterPlus, and Data integration analysis for biomarker discovery using latent components (DIABLO) from the mixOmics R package (version 1.42.0) are among the sophisticated frameworks developed to integrate and interpret these large-scale multi-omics data. These frameworks facilitate cross-platform comparisons and the identification of converging biological signatures in IBD [97].

Moreover, public databases including the Gene Expression Omnibus (GEO), the Human Protein Atlas, and the IBD Multi-omics Database (IBDMDB) are frequently employed to validate findings and perform meta-analyses in IBD [18,98]. The modelling of host–microbiome interactions is also progressing, with the assistance of tools such as MicrobiomeAnalyst and MIMOSA2, which integrate microbial composition with host transcriptomic and metabolomic profiles [99,100]. Emerging technologies, such as spatial transcriptomics and single-cell multi-omics, provide spatially detailed functional mapping of gut tissues, revealing immune infiltration zones, barrier malfunction, and cell–cell interactions during active inflammation [101,102].

While the use of multi-omics methods has significantly improved the understanding of identifying microbial and host markers, a detailed knowledge of the molecular interactions between the host and microbiome remains inadequate. As a result, multi-omics techniques, when combined with systems biology tools that model protein-protein, RNA-RNA, and metabolite-protein interactions, can unravel these complex networks. Nonetheless, the translation of these computational insights into clinical relevance is constrained by obstacles in data integration and a lack of standardization. To ensure biological validity, these *in silico* predictions should be validated by rigorous *in vitro* and *in vivo* experimental studies. These synergistic techniques collectively offer a pathway to improved biomarker identification and the development of targeted, precision therapies for IBD [18].

#### 3.2.6. Software and Databases Supporting Bioinformatics and *In Silico* Approaches for Small Molecule Drug Discovery in IBD

Computational technologies are now essential for advancing research and therapeutic strategies due to the increasing complexity of IBD pathogenesis. A wide variety of technologies currently support *in silico* IBD research, encompassing network construction, docking simulations, and ADMET predictions. These instruments provide extensive analysis of disease progression, compound bioactivity, target identification, and treatment validation. Table 3 outlines frequently used tools within the domains of network pharmacology, cheminformatics, molecular docking, ADMET screening, and molecular dynamics simulations, along with their descriptions, limitations, and citations [90].

The bioinformatics, cheminformatics, and computational tools listed in Table 3 form an integrated workflow system for drug development and disease modelling in IBD. This integrated pipeline is illustrated in Figure 3.

These systems enable researchers to conduct high-throughput screening, gene-target mapping, pathway enrichment, protein-ligand interaction analyses, and predictive ADMET modelling with enhanced efficiency. Among the widely used tools, Cytoscape (3.10.4) is used for constructing and visualizing protein–protein interactions and compound–target interaction networks. Its use with plug-ins such as ClueGO, MCODE, and CytoHubba augments modular analysis, functional annotation, and prioritization of hub genes. Network creation often begins with curated datasets from STRING or STITCH, enabling users to identify empirically confirmed and anticipated connections, thereby establishing the basic biological map for target selection [103,104]. DAVID and KEGG are simultaneously used for interpreting differentially expressed genes, biological processes, tissue distribution, and molecular pathways involved in IBD, including Th17 differentiation, TNF signaling, and epithelial barrier dysfunction. Enrichment analysis through GO and KEGG pathway analysis is fundamental for interpreting transcriptome data in omics-based inflammatory IBD research [130]. R/R studio is well recognized for gene expression, pathway enrichment, and data visualization analysis. For example, various R packages, such as limma, edgeR, clusterProfiler, ggplot2, heatmap, and VennDiagram, can be used to study and analyze pathways and receptors of IBD [119]. OMIM, GeneCards, and DrugBank are commonly utilised for biomarker identification to elucidate disease–gene–drug associations, providing insights into therapeutic repositioning prospects [131].

In cheminformatics and ADMET prediction, SwissADME, pkCSM, and ProTox-II are essential for assessing drug-likeness, pharmacokinetics, and toxicity profiles of small compounds obtained by virtual screening or network pharmacology methodologies [127,128]. Structural modelling tools, including I-TASSER, RCSB PDB, BioLiP, and CastP, facilitate target structure prediction, binding site identification, and ligand–protein interaction analysis. These are often incorporated into downstream docking or simulation workflows, enabling a smooth transition from *in silico* modelling to rational design [121,122,123,124]. Quantum chemical software Gaussian provides a reliable source to calculate the molecular energies and electronic descriptions for quantum chemical analysis and modification of structure. Moreover, GROMACS is a predominant MD simulation tool for dynamic modelling and validating the docking results by simulating ligand and receptor stability in a solvated system over a time setting [132]. Compound databases such as ZINC, ChEMBL, and PubChem provide an extensive source of information on compounds, bioassays, and pharmacological annotations [110,112,114]. Together, the combined use of these tools makes a multidimensional pipeline that goes from analyzing omics data and predicting targets to designing and optimizing small molecules. It helps to understand the pathophysiology of IBD at a systems level and provides a logical way to find new drugs [133].

#### 3.2.7. Artificial Intelligence and Advancement in *In Silico* Techniques in IBD Research

Despite recent therapeutic advancements, the current state of drug discovery for IBD remains inadequate, marked by a limited number of target-specific therapies in the drug development pipeline. Having said that, artificial intelligence (AI) is gradually evolving in IBD research, by offering innovative solutions such as analyzing the vast amount of transcriptomics data, creating networks to understand the role of different targets in IBD. While these approaches are still mostly at the proof-of-concept stage, they have the potential to guide biomarker discovery and support precision medicine [134]. Furthermore, AI-powered tools could change IBD management by enabling more accurate diagnoses, predicting disease severity, and customizing treatment plans for individual patients.

Integrating multi-omics data with AI facilitates the identification of non-invasive biomarkers and also supports the advancement of precision medicine approaches [135]. A significant example of this integration was reported by Sahoo et al., who pioneered an AI-guided discovery pipeline targeted to identify barrier-protective therapeutics for IBD. Their study focused on Boolean implication networks for data-driven target identification, prediction, followed by target validation using organoid co-culture systems and animal models through network-based selection. These synergistic approaches were inspired by the concept of “Phase 0” trial in personalized medicine and demonstrated that AI-guided approaches have the potential to accelerate lead compound discovery [136].

Within the extensive duration of IBD drug discovery, AI can be involved in several phases, including target identification, patient enrolment, trial optimization, and post-market surveillance. Through the automation of data processing and the integration of real-time clinical and molecular data, AI diminishes research timeframes while enhancing the accuracy and repeatability of results. A study by Sedano et al. reported that AI-driven technologies, including machine learning and natural language processing, can be effectively implemented in various aspects of IBD clinical trials. By implementing multi-omics data, AI can refine molecular targets, design trial strategies, and expedite drug development while addressing challenges related to data integration, bias, and ethical considerations [137]. Nonetheless, certain challenges impede the complete incorporation of AI in IBD research. These encompass elevated computing expenses, data standardization challenges, algorithmic transparency, and the necessity for continuous retraining and regulatory validation. 

Moreover, ethical concerns around patient data privacy and algorithmic bias must be managed through rigorous supervision and cooperative governance structures [137]. Given that, the advantages of AI, such as faster drug discovery, reduced healthcare expenses, and enhanced patient outcomes, highlight its potential. By focusing on the ethical implementation, interdisciplinary cooperation, and ongoing validation, AI may complement conventional approaches to transform the clinical management and treatment framework of IBD [137].

#### 3.2.8. Advantages and Limitations of Applying Computational Models in the Early Stage of Drug Discovery

The growing sophistication of computational tools is pointing towards a transition to computer-driven drug discovery, establishing *in silico* methods as an essential tool of the modern drug development pipeline. These *in silico* methods can explore vast chemical spaces for hit identification using high-throughput virtual screening through molecular docking simulation and AI-driven methods and refine these hits into leads with AI-powered potency prediction tools like QSAR. Moreover, *in silico* simulations enhance the rational design of compounds by integrating structural data with predictive analytics, hence augmenting hit-to-lead efficiency and lowering the rate of attrition in preclinical development [138].

However, for *in silico* tools to accomplish this, more robust and better-integrated computational tools should be implemented in the entire drug discovery pipeline. This will ensure a greater impact on the successful progression of initial hits to preclinical and clinical stages [139]. Despite increasing accuracy, computational predictions require validation through vigorous *in vitro* and *in vivo* experimental investigations at each step. Conversely, the experimental data generated from validating these predictions is invaluable for improving the computational models [140]. Expanding training datasets, particularly with data from advanced *in vitro* technologies such as organs-on-a-chip, incorporating functional organoids for better ADMET/PK estimations, reducing reliance on animal studies, and including species-relevant assay data will be key to minimizing the drawbacks of these integrated models. This iterative process of prediction and validation creates a positive feedback system, progressively enhancing the efficiency of *in silico* models so that they can significantly drive compound selection for most drug development objectives [12].

## 4. *In Vitro* Models Using Human Cells in IBD Research

Although *in silico* techniques, AI-driven analytics, and high-throughput omics platforms have transformed the initial stages of drug development and target identification in IBD, experimental validation remains essential. Consequently, the integration of *in vitro* and *ex vivo* human cell-based models, which are easy to use and provide reliable ways to mimic important aspects of IBD pathophysiology, is increasingly needed and beneficial [141]. These models link computational predictions to clinical significance by making it possible to directly study the function of the epithelial barrier, immune responses, host-microbe interactions, and the efficiency of drugs in inflammatory diseases. The following sections examine how various types of 2D monolayer cultures, advanced 3D co-cultures, organ-on-a-chip systems, and stem cell-derived organoids can contribute to a better knowledge of IBD and the development of more targeted and effective drugs [16].

### 4.1. 2D Cell Culture Models (Immunological Monoculture and Co-Culture Models)

#### 4.1.1. Caco-2 (Cancer-Coli-2) Cell Line

Caco-2 cells, derived from human colorectal cancer, are extensively utilized for their capacity to develop into enterocyte-like cells that establish tight junctions. These immortalized human colorectal adenocarcinoma cells secrete key inflammatory mediators, such as IL-6, IL-8, IL-15, TNF-α, and thymic stromal lymphopoietin (TSLP), making them valuable for inflammation-related assays in IBD research [16]. Nonetheless, their monolayer structure lacks the intricacy of the *in vivo* barrier, omitting mucus-secreting goblet cells and non-absorptive lineages. Moreover, their tight junctions are simpler than those observed in impaired IBD settings, restricting their use for barrier dysfunction research [142].

#### 4.1.2. Human Colorectal Adenocarcinoma (HT29) Cell Line

This cancer cell line can be implemented in IBD studies to investigate the activity of TLR receptor, PPAR-γ, pro-inflammatory cytokines, CXCL-1, and COX-2. One of the drawbacks of this cell line study is that it requires an extensive amount of glucose concentration in the medium. So this cell line does not demonstrate a characteristic resemblance to human gut epithelial cells when subjected to a high-glucose medium [16]. Despite this limitation, HT29 cells offer an important model for evaluating immunomodulatory interventions for plant-derived phytochemicals. For instance, Arabinoxylan (AX), a predominant polysaccharide of wheat grain, can be enzymatically hydrolyzed to produce arabinoxylan hydrolyzates (AXH). In a study, AXH has demonstrated to modulate immune responses in HT29 and Caco-2 cells, with higher arabinose substitution in AXH correlating with reduced production of pro-inflammatory cytokines IL-8 and TNF-α, indicating potential anti-inflammatory activity. These findings support the use of AXH as an immunomodulatory agent in colorectal cancer and IBD research [143].

#### 4.1.3. Caco-2/HT29-MTX Cell Line

Combining absorptive Caco-2 and mucus-secreting HT29-MTX cells yields a more physiologically relevant epithelial monolayer. Co-cultures of these two cells can be applied in various studies, such as microbial attachment, gut absorption, permeability through the colonic barrier, and intestinal barrier characteristics [16]. That being said, these co-cultures still lack important things like effective P-glycoproteins and often have open tight junctions that may decrease the integrity of the monolayer [142]. 

Building on this, a study involved an advanced *in vitro* “leaky gut” model by co-culturing Caco-2 and HT29-MTX-E12 cells with differentiated human macrophage-like THP-1 cells or primary monocyte-derived macrophages. This triple-culture system can replicate an inflamed intestine characterized by reduced transepithelial electrical resistance, depletion of tight junction proteins, enhanced permeability to fluorescein isothiocyanate (FITC-dextran), and elevated levels of pro-inflammatory cytokines, including TNF-α, IL-6, and IL-23. Such a model provides a promising pre-clinical method for IBD drug development and offers an exception to *in vivo* testing [144].

#### 4.1.4. T84 Cell Line

This distinctive human adenocarcinoma cell line has been extensively utilized to examine mechanisms related to electrolyte transport, intestinal permeability, and, crucially, signaling pathways of gut inflammatory response in IBD. Nonetheless, the malignant originality and the inadequate epithelium-specific activity exhibited in *in vivo* studies are the main limitations of this T84 cell line [16]. Despite the fact, recent studies on hemp-derived nanovesicles (HNVs) from roots, seeds, sprouts, and leaves further demonstrated significant effects against dextran sulfate sodium (DSS)-induced colitis and liver injury.

*In vitro,* HNVs restored tight and adherent junction proteins and reduced epithelial permeability in DSS-treated T84 cells, while *in vivo* they alleviated gut and liver injury by inhibiting NF-κB activation and oxidative stress. These findings highlight the significance of T84 cells, despite their limitations, as a complementary model for screening anti-inflammatory and barrier-protective therapies in IBD [145].

#### 4.1.5. Peripheral Blood Mononuclear Cells (PBMCs)

PBMCs are composed of dendritic cells, lymphocytes, and monocytes. Polyclonal activators induce the secretion of cytokines from these cells, making them suitable for IBD immune response research. The primary constraint of this method is the existence of phenotypic variations between immune cells and intestinal mucosa cells, even though PBMCs are a convenient source of human immune cells [16]. Recent studies have further demonstrated the utility of PBMCs in evaluating novel drugs. In experimental colitis, for example, it was shown that Mannich curcuminoids have potent anti-inflammatory properties. Curcuminoid suppressed NF-κB activity in a concentration-dependent manner and markedly reduced the expression of important cytokines, such as TNF-α, IL-6, and IL-4, in human PBMCs treated with LPS. These results highlight that PBMC-based assays are significant for evaluating the immunomodulatory potential of novel therapeutic candidates, in addition to aiding in the understanding of immunological dysregulation in IBD [146].

#### 4.1.6. Human Acute Monocytic Leukemia Cell Line (THP-1)

THP-1 cell lines are among the most promising human leukemia cells obtained from real patient cells. During differentiation, these cells resemble primary monocytes and macrophage cells. This cell line is mainly used for research involving reactive oxygen species (ROS) generation and inflammatory mediation. Due to the homogeneous genetic background, this cell line has high reproducibility and minimal variability of cellular phenotype. Therefore, this cell line is more suitable than the previously discussed PBMC one and any other macrophage cell line. Moreover, their benefits outweigh the PBMC cell line, which is that the siRNA of these cells allows the convenient downregulation of protein expression. Thus, these cells are now particularly prioritized over other cell line assays due to their high vitality, longer shelf life, and, most importantly, the capability of mimicking IBD-related inflammation properly [16].

THP-1 cells have also been approved for long-term cytotoxicity and genotoxicity testing in addition to immunological studies. In a study, a 14-day repeated-dose assay on novel candidates revealed that Paraquat caused significant cytotoxicity and DNA damage, whereas other compounds, such as 3-Nitropropanoic acid (3-NPA), had moderate effects. These characteristics make THP-1 a strong and adaptable *in vitro* model for studying inflammation and evaluating the long-term safety of new candidate drugs [147].

#### 4.1.7. RAW 246.7 Cell Line

The RAW 246.7 cell line, obtained from a mouse tumour using the Abelson leukemia virus, has also revealed that these cells can significantly simulate the gut environment and can successfully mimic TLRs, which play a pivotal role in IBD pathogenesis. The RAW 246.7 cell line can simulate host-microbiome interactions, which is one of the key pathologies in IBD. Significantly, upon LPS stimulation, RAW 264.7 macrophages polarize into proinflammatory M1-like cells, characterized by the expression of CD86, IL-6, TNF-α, IL-1β, and CCL2. In contrast, M2-like cells exhibit anti-inflammatory markers, including CD206, TGF-β, and Arg1.

In a study, treatment with a tetracyclic quinolizidine alkaloid (Matrine) demonstrated a reduction in CD86 expression and proinflammatory cytokines, while simultaneously augmenting M2 markers and fostering an anti-inflammatory phenotype. This highlights the importance of RAW 264.7 cells in studying macrophage polarization and evaluating novel IBD treatments [148]. Having said that, when utilizing RAW 246.7 cells, caution should be implemented during data interpretation, as the result can be altered during continuous culture. Additionally, if human and mouse cells are merged (co-culture of Caco-2/RAW 246.7), different cross-reactions should be reconsidered [16].

### 4.2. Ex Vivo Three-Dimensional (3D) Culture Model

Considering the drawbacks of conventional 2D cell culture systems, *ex vivo* and 3D culture models have emerged as more physiologically significant platforms for modelling the etiology of IBD. *Ex vivo* and 3D models can provide a more insightful representation of gut pathophysiology to study intestinal inflammation and can correlate with the cell line and animal study results. Notably, these models serve as a vital link between cell-based assays and *in vivo* studies, thereby enhancing the translational significance of preclinical research in IBD [142]. This section provides a summary of the current 3D culture models employed in IBD research, emphasizing their applications, benefits, and inherent limitations.

#### 4.2.1. Organ-on-a-Chip (OoC) Model

Organ-on-a-chip (OoC) technology has attracted considerable interest as a cutting-edge method for investigating intestinal inflammation and drug delivery strategies in the field of IBD research. By integrating microfluidic technologies and human-derived cells, OoC platforms can effectively mimic the dynamic interactions among epithelial, endothelial, and immunological components of the gut more accurately than any other 2D culture or animal model [149]. For example, Tataru et al. developed a cytokine-induced inflammatory model utilizing co-cultured Caco-2 and HUVEC cells subjected to TNF-α and IFN-γ exposure. This model effectively replicated cytokine and chemokine release associated with IBD-related barrier disruption, facilitating the assessment of anti-inflammatory drugs [150]. Similarly, Kaden et al. designed an organ-on-a-chip model to explore host-microbiota interactions and evaluate precision-targeted therapeutics. In contrast, Beaurivage et al. presented a gut-on-a-chip system that integrates intestinal epithelial cells and macrophages to replicate *in vivo* immunological responses.

Their platforms exhibited distinct inflammatory indicators, including IL-8, CXCL10, and CCL-20, which closely correlated with the transcriptional patterns identified in IBD patients [151,152]. A further advancement of this innovative microfluidic system has been proposed by Liu et al. by integrating oxygen gradient control into the chip architecture. This gut-on-a-chip apparatus selectively regulates oxygen transport via microchannel architecture, establishing hypoxic conditions that mimic intestinal physiology. In the presence of inflammatory stimuli such as TNF-α and lipopolysaccharide, the probiotic *Bifidobacterium bifidum* substantially retained barrier integrity and reduced tissue damage, demonstrating its protective activity in the context of IBD [153]. Furthermore, Beaurivage and colleagues employed a high-throughput OrganoPlate-based gut-on-a-chip model to replicate barrier disruption and cytokine synthesis.

By employing on-chip knockdown of critical inflammatory mediators RELA and MYD88, they effectively modulated the inflammatory phenotype, establishing a viable model for drug development and mechanistic studies [154]. The OoC model, as outlined by Shin et al., offers a more human-based and less variable *in vitro* model for IBD, which can evolve into a robust preclinical model.

Despite their potential, these organ-on-a-chip models possess certain limitations. For instance, unpredictable responses to inflammatory stimuli are prone to the major drawbacks that may raise safety issues in clinical applications. Nevertheless, the ongoing advancement of these systems, primarily through the incorporation of patient-derived epithelial, immunological, and microbial elements, possesses significant potential for precise modelling of IBD and the generation of targeted, personalized medicine [155].

#### 4.2.2. 3D Tissue Culture, Stem Cells, and Organoids Design

In addition to OoC models, 3D tissue cultures, organoids, and stem-cell-derived models have become indispensable components in the research field of IBD. In comparison to conventional monolayer cultures, these platforms provide a more comprehensive replication of the intestinal epithelium’s functional and compositional characteristics [156]. Intestinal organoids, originating from mature or induced pluripotent stem cells, can mimic the fundamental features of the gut, encompassing the crypt-villus architecture, epithelial barrier integrity, and host-microbe interactions [157]. They have been widely utilized to examine the pathological role in IBD, encompassing epithelial injury, immunological dysregulation, and microbial imbalance. Moreover, their predictive capability makes them appropriate for screening regenerative medicine in diseased conditions [158,159]. Given that, these models have certain limitations, such as a lack of skills and the expense of materials and equipment, which can limit their widespread application [142]. Furthermore, as the models primarily feature epithelial cells, they often lack crucial non-epithelial immune and stromal cells, gut-microbial models that may contribute to the intestinal environment. Moreover, maintaining the models for an extended period to replicate the chronic conditions remains a significant challenge [160].

Also, the dissimilarities of different organoids derived from many individuals may contradict the standardization and reproducibility of research outcomes and drug discovery efforts [161]. Nevertheless, the prospects for 3D tissue culture in IBD research are optimistic. Incorporating immunological and microbial elements into these models will augment their translational significance. Patient-derived organoids possess the capacity to further personalize treatment approaches by detecting molecular subtypes and enabling targeted therapeutic interventions [162]. Recent advances have reported the development of an epithelial inflammatory injury model using intestinal organoid cultures. In this model, exposure to cytokines such as TNF-α, IL-1β, and IFN-γ mimics the epithelial injury patterns seen in IBD. These models provide a system for tracking and dissecting epithelial repair pathways. It also provides a method for screening anti-inflammatory drugs, thereby bridging the gap between simplified monolayers and patient biopsies [163]. Another study also reported that organoid biobanks are now being used with clinical cohorts to inform stratification in early interventional studies [164]. Ultimately, these models may accelerate the transition from laboratory results to clinical application through preclinical “Phase 0” investigations using microbiota-targeted medicines designed to restore intestinal homeostasis in IBD [165].

#### 4.2.3. Single-Cell RNA Sequencing (scRNA-Seq) and Whole-Genome Sequencing (WGS) in IBD Research

Advanced 3D cultures and patient-derived organoids offer physiologically relevant models for investigating IBD; however, their full potential is further enhanced when combined with high-resolution genetic profiling and multi-omics tools. The use of scRNA-seq and WGS has become an important tool for identifying the cellular complexity and genetic framework associated with IBD pathogenesis. When integrated with 3D models, these tools provide unparalleled insight into cell-type-specific responses, genetic determinants, and signaling pathways within a near-physiological setting [162]. scRNA-seq enables researchers to analyze cellular heterogeneity, immune cell dynamics, and epithelial dysfunction at the granular level by providing a detailed transcriptional map of individual cells within the intestinal mucosa. Employing both droplet-based (high-throughput) and plate-based (high-resolution) methodologies, scRNA-seq has elucidated critical markers in IBD, notably the elevation of proinflammatory cytokines and a transition from IgA to IgG-producing B cells with advanced disease progression [166]. Moreover, based on the study’s findings, scRNA can be used to understand host-microbe interactions, as it may reveal that patients with high levels of bacteroides demonstrate reduced inflammation accompanied by elevated interferon signaling [167]. Most importantly, the scRNA-seq method can be applied to discover novel targets such as Triggering Receptor Expressed on Myeloid cells 1 (TREM1), which is involved in both UC and CD pathology. Consequently, a novel monoclonal TREM-1 inhibitor antibody, CEL383, is under clinical trial now, demonstrating the significance of early scRNA-seq data in translational research. However, most of these candidates are still undergoing preclinical validation rather than being established drug targets and more robust experimental analysis is further needed [168].

In parallel, WGS contributes to IBD research by identifying effector genes involved in IBD pathology, such as the RASGRP1 gene in tissue-resident memory T cells, a key regulator of CD susceptibility. This identification can highlight the significance of cell-type biomarker expression in IBD pathophysiology and guide future biomarker discovery research [169]. The combination of scRNA-seq with genome-wide association studies (GWAS) has enabled a more accurate detection of disease-associated polymorphisms in functional cellular settings. The IBDverse program, which examined over 2.4 million transcriptomes, associated genetic risk loci with effector genes in 52% of instances, more than twice the previous resolution. Tools such as scGWAS and scDRS enable researchers to determine genetically significant cell groups and treatment targets [170]. These tools are expanding the accuracy of IBD research, but widespread applications of these tools in therapeutic pipelines are still at an early stage [171].

Undoubtedly, high cost, technical complexity, and large sample size requiring statistical excellence are the most challenging factors in these studies [172]. Furthermore, the standardization of methods for integrating these omics technologies with organoid platforms and *ex vivo* models is still ongoing. As bioinformatics advances and sequencing costs decrease, the combined use of 3D organoid systems with scRNA-seq and WGS will likely be pivotal in modelling disease heterogeneity, identifying predictive biomarkers, and accelerating targeted drug development in Inflammatory Bowel Disease [173,174].

## 5. Animal Models in IBD: Connecting *In Vitro* Insights with *In Vivo* Outcomes

While advanced *in vitro* and *ex vivo* platforms, such as organoids and organ-on-a-chip (OoC) models, have significantly broadened our understanding of IBD at the cellular and tissue levels, they are still inherently limited in their ability to replicate the diverse, systemic physiology of a living organism. To bridge this critical translational gap, animal models remain important tools in IBD research, as they help to study immune modulation, disease etiology, and pharmacodynamic responses [175]. The controlled biological environment provided by animal models enables the replication of various IBD-related phenomena, including mucosal injury, immune dysregulation, cytokine inducement, and microbiota-mediated inflammation. To date, more than 60 distinct models are available for studying IBD inflammation, such as chemically induced colitis, adoptive T-cell transfer, spontaneous mutations, genetically engineered mice, and microbiome-influenced models. Together, they enhance our ability to validate therapeutic hypotheses that were derived from earlier *in silico*, *in vitro*, and *ex vivo* findings, while each provides unique insights [176]. Notably, recent developments have highlighted the synergistic integration of human-derived gut organoids into colitis animal models. For instance, the transplantation of human induced pluripotent stem cell-derived intestinal organoids (hiPSC-IOs) into mice with Dextran Sodium Sulfate (DSS) induced colitis resulted in substantial reductions in epithelial damage, reduced production of pro-inflammatory cytokines, and the restoration of epithelial barrier function. The engrafted organoids effectively produced tight junction proteins (e.g., ZO-1, occludin) and showed the ability to differentiate into mature intestinal cell types, providing a regenerative strategy for IBD therapy [177]. Similarly, Sugimoto et al. demonstrated that implanted organoids could grow, integrate, and survive for an extended period in the colonic mucosa. This shows that these humanized animal models could be used as a preclinical platform [178].

These combinatorial models are particularly effective for evaluating personalized medicines and exploring host-microbiome-epithelium interactions in a therapeutically applicable setting in IBD drug discovery. Even today, the problems with animal models, such as species- and strain-specific models, the lack of human immune-derived components from humans, and the fact that they do not always accurately predict illness, highlight the importance of using models from multiple types [179]. Table 4 outlines prominent murine IBD models, emphasizing their induction techniques, pathophysiological characteristics, advantages, and limitations.

As indicated in Table 4 each model has its unique advantages and limitations compared to the others [191]. Among these, the DSS model is mostly used due to its repeatability and cost-effectiveness, accurately simulating the epithelial barrier degradation observed in UC [175]. Conversely, TNBS and oxazolone-induced colitis are more appropriate for assessing Th-1 and Th-2 type inflammatory responses, respectively [182]. Immunological and genetically engineered models, such as the CD4^+^CD45RB^high^ T cell transfer model and IL-10 mutant mice, elucidate chronic inflammation and immunological dysregulation associated with CD and UC [186,192]. Given that, no one particular animal model can adequately mimic the complex pathophysiology of IBD [191]. Each model has its own set of problems, such as the duration of the disease, the composition of the immune cells, and responses to human biologics. As seen in Table 4 combining data from different models enhances mechanistic understanding and strengthens translational significance. Additionally, data from prior *in silico* investigations, such as molecular docking, network pharmacology, and gene set enrichment analysis, may enhance target selection for animal validation. For example, pathways like TLR4/MyD88/NF-κB, identified through computational modelling, can be empirically confirmed in DSS or TNBS models, facilitating the repeated refinement of mechanistic hypotheses and treatment options [193].

## 6. Integrating Computational, Cell Culture, and Animal Models: Approaches to Overcome Limitations of Conventional Drug Discovery

Conventional drug development is a prolonged, costly, and high-risk process, particularly for complicated diseases such as IBD, which involves intricate interactions among immunological dysregulation, genetic predisposition, microbial dysbiosis, and environmental influences. The pipeline from target selection to preclinical assessment and clinical validation is often hindered by inadequate translational predictability, elevated attrition rates in late-stage studies, and a limited understanding of systems-level drug responses [194]. A significant amount of investment is required, and failures, particularly in Phase-III or Phase-IV due to high toxicity or lack of efficacy, are time-consuming and costly. Furthermore, adverse effects arise from unforeseen interactions with off-targets and downstream biological processes, and inadequate pharmacokinetics frequently compromise therapeutic candidates that initially show potential in preliminary assessments.

New approaches are being developed to address these issues by focusing on target selection within the larger context of molecular networks and host-system interactions. Network pharmacology, cheminformatics, and virtual screening are examples of *in silico* methods that can be essential in this context. They allow researchers to examine large volumes of data and study the relationships between drugs, targets, and pathways before commencing laboratory research.Despite the success of network-based pathway analysis in drug discovery, several pressing issues pose significant challenges, including conflicting and questionable database quality, bias in computational tools towards well-studied entities, poor data quality, a lack of standardization in evolving technologies, and the inability of static computational models to represent dynamic biological systems [194].

While molecular docking has excellent potential for ligand and structure-based drug discovery, poor conformational sampling along with false scoring often restricts its application, typically requiring target crystal structures. Nonetheless, researchers are now employing homology models, combining molecular dynamics for optimization and dynamic analysis, and benefiting from the increasing availability of experimental structures to surpass these scoring methods. Advanced post-docking refinement helps to improve virtual screening success rates even further [195]. Considering the poor ADMET properties of experimental drugs in the last stages of the drug-discovery process, the pharmaceutical industry is shifting to a “fail early, fail cheap” strategy. This involves prioritizing early assessment and optimization of ADMET characteristics to reduce attrition [196]. While *in vitro* and *in vivo* methods are valuable, overall cost and complexity limit their use for screening many early-stage compounds. The significance of merging computational, cell line, and animal models is herein elucidated.

Each system possesses unique advantages: *in silico* models facilitate hypothesis generation and expedited screening; *in vitro* systems (including 2D and 3D cultures, co-cultures, and organoids) provide mechanistic insights within controlled environments; and *in vivo* models elucidate systemic responses, pharmacokinetics, and host–microbiome–immune interactions. When properly integrated, these platforms provide a modern and adaptive framework. Computational predictions can guide experimental design, while *in vitro* and *in vivo* results can refine algorithmic models and contribute to data-driven simulations. Such integrative workflows are especially valuable in IBD research, where disease heterogeneity and tissue-specific pathology demand context-aware validation of therapeutic candidates. Figure 4 illustrates the synergistic role of integrated computational, cell line, and animal models in accelerating IBD drug development, highlighting a systems-based translational approach that enhances target predictability and improves therapeutic outcomes.

This schematic illustrates a drug development framework that integrates *in silico* screening, *in vitro* validation, and *in vivo* assessment for IBD. For reproducibility, AI-assisted simulations should select candidate compounds and predicted molecular targets, which are screened in disease-relevant models such as 2D/3D cultures, co-cultures, and gut organoids. Only molecules demonstrating efficacy in restoring epithelial integrity, regulating cytokine responses, or modulating microbiome composition can be advanced into *in vivo* IBD models to assess systemic pharmacokinetics and immune responses. Afterwards, the data correlation is reassessed, where data validated *in vitro* and *in vivo* outcomes are fed back into the computational platform, closing the feedback loop and enabling more accurate and informed molecular redesign. This iterative refinement strategy may reduce failure rates by filtering out non-viable compounds early while continuously improving mechanistic insight. Notably, there are several studies that have reported the impact of integrated workflows. For example, JAK1 inhibitor Filgotinib was approved for UC treatment optimized via a combination of virtual kinase selectivity profiling, cytokine suppression in co-cultured T-cell/epithelial systems, and targeted efficacy testing in murine colitis models [197]. Furthermore, another study by Greenhalgh et al. outlined that a combined *in vitro* and *in vivo* strategy could represent the molecular impacts of a synbiotic regimen on colorectal cancer-derived cells. This study demonstrated how computational predictions integrated with disease-relevant cellular models can mechanistically guide therapeutic strategies and enhance translational relevance [198]. Therefore, in light of these experimental evidence, these integrative workflows offer a robust platform for accelerating IBD drug discovery, while also improving prediction accuracy and translational potential.

## 7. Conclusions and Future Direction

IBD poses significant challenges to the healthcare system due to the complex pathology combining genetic, microbial, and immunological factors. Unfortunately, current treatment protocols available for IBD target temporary relief of patient symptoms but do not target the molecular basis of the disease. There is an urgent need to find more innovative, efficient, and safer small-molecule drugs to target IBD more effectively with the shortest possible time. Modern drug discovery approaches, including *in vivo* molecular docking, network pharmacology, computational modelling, bioinformatics, and artificial intelligence, are gaining popularity as effective methods in modern drug discovery. However, to validate the pathway analysis by bioinformatics and *in silico* studies, advanced *in vitro* models using human cells, patient-derived tissues, organs, and gut-on-a-chip approaches can provide more human-relevant models rather than traditional cell line assays. When these methods are integrated with high-throughput techniques such as scRNA-seq and network-based pathway analysis, they become an invaluable method for identifying the complex pathophysiology of the disease. Animal models have also proven critical in understanding disease progression and for validating targets identified through computational and cell line studies. Nevertheless, until today, a significant translational gap remains between human physiology and animal models. As a result, the synergistic application of *in silico* predictions, validated by complex *in vitro* investigations and then confirmed by selected *in vivo* models, is one of the most promising techniques for identifying targets and developing new medicines.

Future research should prioritize the integration of computational and experimental methods in the following ways. Firstly, the integration of network pharmacology, docking, and AI-driven models in the same pipeline of organoid and gut-on-chip technologies to encompass disease heterogeneity. Secondly, using the most physiologically relevant *in vivo* models to validate the *in silico* and *in vitro* findings. Thirdly, implementing ethical frameworks for AI-driven drug discovery to overcome bias and ensure robustness. Finally, by integrating single-cell and multi-omics data with patient-derived experimental model data to cover disease complexity and identify therapeutic targets. By methodological advancements in these strategies, it is expected to expedite the overall IBD drug discovery pipeline. This will lead future IBD research shifting from symptomatic relief to mechanistically informed, personalized medicines that can ultimately transform clinical care.

## Figures and Tables

**Figure 1 pharmaceuticals-18-01536-f001:**
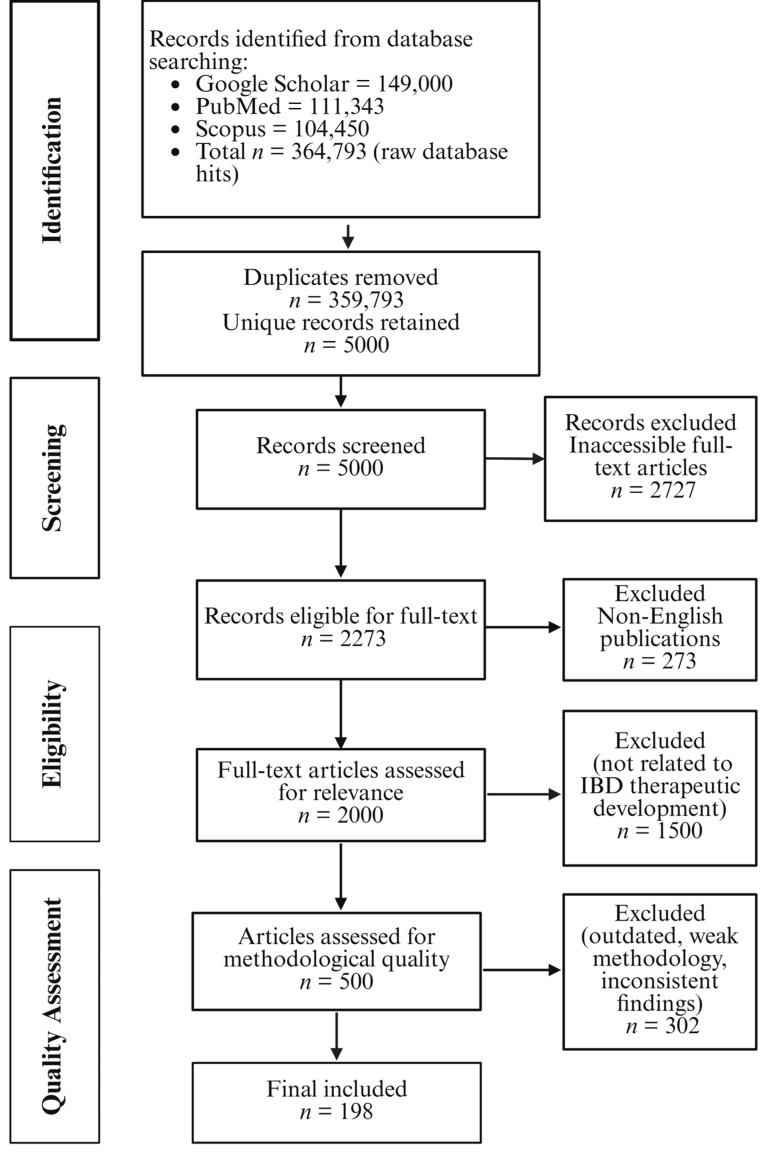
PRISMA flow diagram illustrating the selection process of studies included in this review. (Created in https://BioRender.com, Smout, M. (2025) https://app.biorender.com/illustrations/68d710f57a2fce14991b0ba8?slideId=b9e5fc06-e399-4332-b97b-dfc8ce5364cf.

**Figure 2 pharmaceuticals-18-01536-f002:**
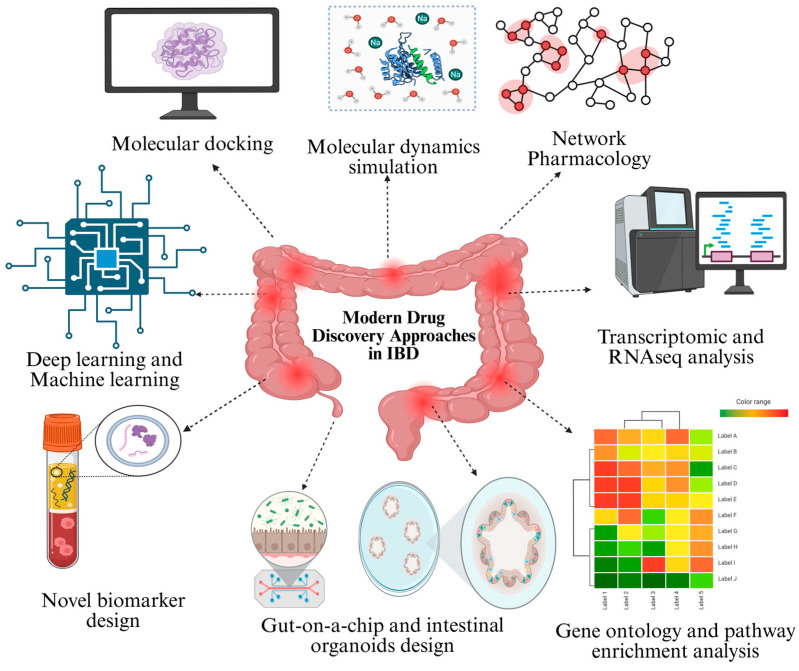
Modern Drug Discovery Approaches for Inflammatory Bowel Disease. Integrated *in vivo* methods (molecular docking, molecular dynamics, network pharmacology, machine learning, transcriptomics) and experimental platforms (biomarker design, organoids, gut-on-a-chip) provide complementary strategies for target identification and therapeutic evaluation in IBD. (Created in https://BioRender.com, Smout, M. (2025) https://app.biorender.com/illustrations/650b05b594442a147a820856?slideId=62648d43-a1e9-426b-b247-af65bf6dda59.

**Figure 3 pharmaceuticals-18-01536-f003:**
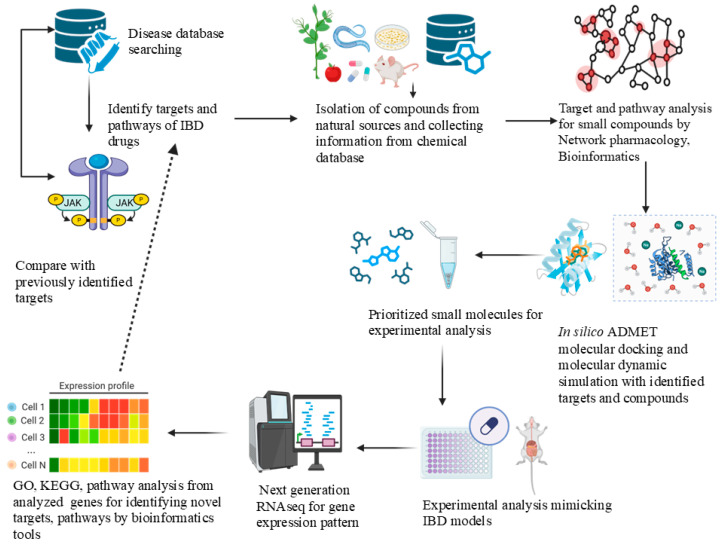
Integrated workflow for small molecule drug discovery in inflammatory bowel disease (IBD). The pipeline integrates compound isolation, database mining, network pharmacology, along with *in silico* ADMET, molecular docking, and molecular dynamics. Selected candidates are subjected to experimental validation in IBD models and transcriptome analysis (RNA-seq, GO, KEGG). This process facilitates the discovery of innovative therapeutic targets for IBD, building upon previously reported ones. (Created in https://BioRender.com, Smout, M. (2025) https://app.biorender.com/illustrations/650b05b594442a147a820856?slideId=069b47e9-17c0-4eaa-923d-de80452b123b.

**Figure 4 pharmaceuticals-18-01536-f004:**
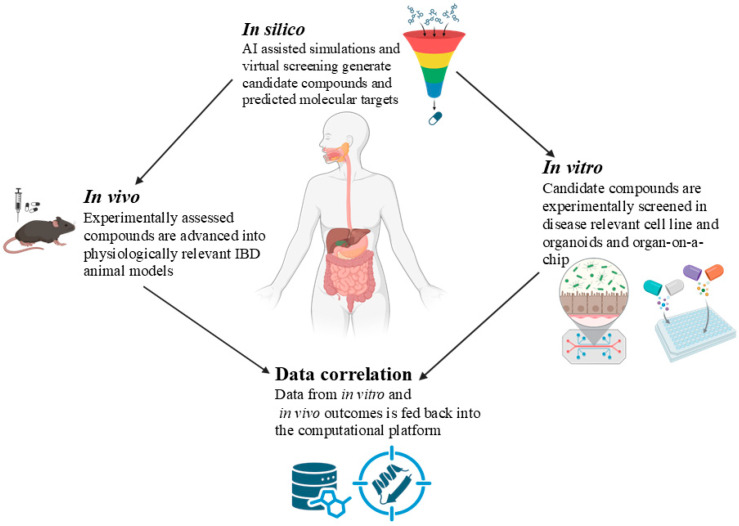
Schematic depiction of an AI-driven feedback loop connecting computational predictions, experimental models, and animal validation for IBD drug discovery. Candidate drugs are developed *in vivo*, screened in human-relevant *in vitro* systems, validated in *in vivo* models, and the outcomes of the experimental results fed back into the computational system. This closes the feedback loop to refine each successive iteration. (Created in https://BioRender.com, Smout, M. (2025) https://app.biorender.com/illustrations/650b05b594442a147a820856?slideId=5e6496ba-df34-45fc-afdb-8fbca8fe102c.

**Table 1 pharmaceuticals-18-01536-t001:** Current and Emerging Biomarkers in IBD Research.

Category	Biomarker Type	Indicators	Source	Application	Validation Status
Diagnostic	Metabolomic [27,28]	Short-chain fatty acids (SCFA), bile acids, sphingolipids, tryptophan metabolites	Fecal sample	Non-invasive diagnosis, disease severity	Clinical use
	Proteomic [29,30]	Calprotectin, Lactoferrin, Calgranulin C, lipocalin-2. Blood:	Fecal and blood samples.	Disease severity assessment and non-invasive diagnosis.	Clinical use
	MicroRNA (miRNAs) [31,32]	Tissue: miR-192, miR-375, and miR-422b. Serum: miR-146a, miR-146b, miR-320a, miR-126, and let-7c.	Tissue and blood samples.	Differentiation of IBD subtypes (UC and CD)	Research/Early validation
	Inflammatory Biomarker [25]	Blood: CRP, ESR, ASCA, LRG, LCN-2 Feces: Fecal calprotectin, Lactoferrin, S100A12	Blood, Fecal samples	Diagnosis, Subtype differentiation	Clinical use/Early Validation
Predictive/Prognostic	Transcriptomic [33]	Ileal: FOLH1, CA2 Colonic: REG3α	Mucosal biopsies (ileum/colon)	Predicting histological and endoscopic healing	Research/Early validation
	Serological [34]	CPa9-HNE	Serum	Assessment of endoscopic activity and neutrophil infiltration.	Research/Early validation
	Genetic [35]	RUNX1	Colon tissue	IBD progression, risk of colorectal cancer.	Experimental/Preclinical
Monitoring	Gut Microbial Biomarkers [36,37]	CRP, Fecal calprotectin (FC), Calgranulin-C (S100A12), Stool lactoferrin (SL)	Stool samples	Monitoring of mucosal healing, guidance for treatment decisions.	Clinical use
	Myeloperoxidase [38]	Myeloperoxidase (MPO)	Fecal samples	Monitoring of neutrophilic inflammation and disease severity	Experimental/ Preclinical
	Transcriptomic [33]	Colonic: REG3α	Colon biopsies (ileum/colon)	Inflammatory burden, mucosal repair	Research/ Early validation

**Table 2 pharmaceuticals-18-01536-t002:** Common Computational Tools for Molecular Docking, Visualization, and Simulation in Small-Molecule Drug Discovery.

Category	Tool	Algorithm/Core Approach	Key Features	Accessibility	Limitations
Docking Engines	AutoDock [55,56]	Lamarckian genetic algorithm	Widely used, open-source, supports HTS	https://autodock.scripps.edu/ (accessed on 1 August 2025)	Limited protein flexibility, simplified scoring
	SwissDock [57]	EADock DSS (evolutionary algorithm + local search)	Web-based Protein-ligand docking	http://www.swissdock.ch/ (accessed on 1 August 2025)	Limited to small molecules only.
	ZDOCK/M-ZDOCK [58,59]	Fast Fourier Transform (FFT) based correlation	Protein–protein docking.	https://zdock.wenglab.org/ (accessed on 1 August 2025)	Requires post-docking refinement.
	MedusaDock [60]	Monte Carlo-based docking	Flexible ligand docking.	https://dokhlab.med.psu.edu/cpi/ (accessed on 1 August 2025)	Limited to rotamer-based flexibility.
	LeDock [61]	Hybrid of simulated annealing + Genetic algorithm	High accuracy for small molecule docking.	https://www.lep har.com/software (accessed on 1 August 2025)	Time-consuming per ligand.
	MOLS 2.0 [62]	Mean-field optimization with scoring functions	Peptide and protein-ligand docking.	https://sourceforge.net/projects/mols2-0/ (accessed on 1 August 2025)	Docking accuracy needs improvement
	rDOCK [63]	Genetic + Monte Carlo search	Ligand and nucleic acid docking	https://rdock.github.io/ (accessed on 1 August 2025)	Linux-only
	HADDOCK [64,65]	Restraint-driven docking (ambiguous interaction restraints)	Integrates experimental data, protein–protein/nucleic acid docking	https://rascar.science.uu.nl/haddock2.4/ (accessed on 1 August 2025)	Requires experimental input; rigid assumptions
	ClusPro [59]	FFT-based+ Clustering	High-accuracy protein–protein docking	https://cluspro.org/help.php (accessed on 1 August 2025)	Relies on structural conformity
	Glide (Schrödinger) [66]	Hierarchical systematic search with scoring filters	Widely used for ligand-receptor docking.	https://www.schrodinger.com/ (accessed on 1 August 2025)	Poor results with bulky ligands.
	GOLD Dock [67]	Genetic algorithm	High receptor flexibility supports	https://www.ccdc.cam.ac.uk/solutions/software/gold/ (accessed on 1 August 2025)	Biased towards specific docking.
Integrated Suites/Platforms	Biovia Discovery Studio [68]	Contains CDOCKER (CHARMm SA), LibDock (hotspot-based)	Protein prep, ligand interaction analysis	https://discover.3ds.com/discovery-studio-visualizer-download (accessed on 1 August 2025)	Cannot perform docking calculations.
	Molecular operating environment [69]	Lamarckian Genetic Algorithm	Structure and pharmacophore docking	https://www.chemcomp.com/en/Products.htm (accessed on 1 August 2025)	Commercial licensing required.
Visualization and Modeling Tools	UCSF Chimera [70]	Visualization and external tool integration (not docking)	Ligand interaction analysis	https://www.cgl.ucsf.edu/chimera/ (accessed on 1 August 2025)	Does not perform docking.
	Swiss-PdbViewer [71]	Genetic algorithm	Homology modelling and minimization	https://swissmodel.expasy.org/ https://spdbv.unil.ch/ (accessed on 1 August 2025)	One PDB at a time.
	PyMOL [72]	Molecular visualization (not docking)	Widely used for structural visualization and interaction analysis	https://www.pymol.org/ (accessed on 1 August 2025)	Limited features in the academic version.

**Table 3 pharmaceuticals-18-01536-t003:** Key Bioinformatics, Network Pharmacology, and Computational Tools Used in IBD Research.

Tool/Database	Application	Limitations	Website
STRING [90,103]	PPI network construction	Limited Gene Ontology classification	https://string-db.org/ (accessed on 3 August 2025)
STITCH [104]	Protein–small molecule interactions	Limited to known (430,000) Compounds	http://stitch.embl.de/ (accessed on 3 August 2025)
HMDB [105]	Human metabolite information.	Incomplete spectral data of many known compounds.	https://hmdb.ca/ (accessed on 3 August 2025)
MiMeDB [106]	Microbiome-derived small molecules	Requires frequent updates	https://mimedb.org/ (accessed on 3 August 2025)
HIT 2.0 [107]	Herb-target-compound data	Focused on herbal compounds only	http://hit2.badd-cao.net/ (accessed on 3 August 2025)
OMIM [108]	Human genes and phenotypes	Mainly Mendelian inheritance disorders	http://www.ncbi.nlm.nih.gov/omim/ (accessed on 3 August 2025)
GeneCards [109]	Gene-centric data from more than 200 sources	Human-only; no cross-species data is available.	http://www.genecards.org/ (accessed on 3 August 2025)
PubChem [110]	Largest compounds database on chemical and bioassay information.	Limited experimental protocol of bioassay data.	https://pubchem.ncbi.nlm.nih.gov/ (accessed on 3 August 2025)
UniProt [111]	Protein sequence, annotation, functional and structural information.	Lack of information about enzyme activity in biochemical reactions.	http://www.uniprot.org/ (accessed on 3 August 2025)
ChEMBL [112]	Drug–target activity data.	Limited information for novel compounds and targets.	https://www.ebi.ac.uk/chembl (accessed on 3 August 2025)
DrugBank [113]	Drug profiles, mechanism of action and pharmacokinetics.	May lacks information about investigational and experimental drugs.	https://go.drugbank.com/ (accessed on 3 August 2025)
ZINC [114]	Commercial small-molecule database.	Lacks full pharmacophore data.	https://zinc20.docking.org/ (accessed on 3 August 2025)
CytoScape [115]	Network visualization and analysis.	Memory-intensive.	https://cytoscape.org/ (accessed on 3 August 2025)
KEGG [116]	Pathway enrichment and maps analysis.	Depends on published articles, manually curated; slower updates.	https://www.kegg.jp/ (accessed on 3 August 2025)
DAVID [117,118]	Gene enrichment and annotation.	Sensitivity varies across background gene lists, scoring systems, tests.	https://davidbioinformatics.nih.gov/ (accessed on 3 August 2025)
R/RStudio [119]	Statistical and pathway analysis.	Requires coding knowledge, statistics and data analysis.	https://cran.r-project.org/ (accessed on 3 August 2025)
RCSB PDB [120]	Protein structure database	Some PDBx/mmCIF entries only files are not readable by docking tools.	https://www.rcsb.org/ (accessed on 3 August 2025)
BindingDB [120]	Protein–ligand affinity data.	Focuses binding affinities over other kinetics, allosteric effects, and complex structural information.	http://www.bindingdb.org/ (accessed on 3 August 2025)
BioLiP2 [121]	Protein-ligand interaction information.	Lacks RNA-ligand interaction application.	https://zhanggroup.org/BioLiP (accessed on 3 August 2025)
CastP [122]	Predicts protein binding sites.	Limited to annotated PDBs only.	http://sts.bioe.uic.edu/castp (accessed on 3 August 2025)
I-TASSER [123,124]	Protein 3D structure prediction, homology modelling	Accuracy depends on the templates of the protein.	https://zhanggroup.org/I-TASSER/ (accessed on 3 August 2025)
Gaussian [125]	Quantum energy calculations of small molecules.	High computational cost.	https://gaussian.com/ (accessed on 3 August 2025)
SwissADME [126]	ADME and drug-likeness prediction of small molecules.	Peptides not supported with SwissADME.	http://www.swissadme.ch/ (accessed on 3 August 2025)
pkCSM [127]	ADMET property prediction	Experimental validation needed.	https://biosig.lab.uq.edu.au/pkcsm/ (accessed on 3 August 2025)
ProTOX II [128]	Toxicity prediction models of small molecules.	Dependency on 2D input may lack the information of 3D structures.	http://tox.charite.de/protox_II (accessed on 3 August 2025)
GROMACS [129]	Molecular dynamics simulation.	Limited for large molecular weight protein.	https://www.gromacs.org/ (accessed on 3 August 2025)

**Table 4 pharmaceuticals-18-01536-t004:** Prominent murine models used in IBD research.

Model Type	Induction Method	Disease Features	Advantages	Disadvantages
Dextran Sodium Sulfate (DSS) [175,180]	3–5% DSS (C57BL/6 or BALB/c mice); 4% DSS in SD rats.	Ulceration, submucosal edema, bloody diarrhea	High reproducibility mimics human UC, cost-effective.	Excess dosing causes severe inflammation; lacks chronicity
TNBS (Trinitrobenzene Sulfonic Acid) [181,182,183]	Intrarectal TNBS (BALB/c mice, Wistar rats)	Bloody diarrhea, weight loss, IL-12 driven colitis	Suitable for anti-TNF-α drugs and conventional therapies.	Species-specific susceptibility; limited human relevance.
Oxazolone [183,184]	1% oxazolone in BALB/c, SJL/J strains	Weight loss, goblet cell depletion, IL-9 upregulation	Models IL-4R and IL-9 pathways relevant in UC	Ineffective in C57BL/6 strain; short-lived inflammation
Acetic Acid-Induced Colitis [183,185]	4% acetic acid in SD rats (1.5 mL)	Diarrhea, rectal bleeding, epithelial injury	Simple, rapid, cost-effective	Severe epithelial damage at high dose; poor chronic resemblance.
Adoptive T-cell Transfer [183,185]	CD4^+^CD45RB^high^ T cells into Rag1 KO mice	Chronic colitis, loose stool, immune-driven inflammation	Insight into T-cell-mediated pathology	Requires an immune-deficient host; time-consuming
IL-10 Knockout [186,187]	IL-10 null C57BL/6 mice	Chronic inflammation, bowel thickening	Useful for understanding IL-10 signalling	Symptom onset variable; requires close monitoring
SAMP1/YitFc (Spontaneous) [188]	Genetically selected AKR/J mice	Ileitis, perianal fistulae, chronic lesions	Spontaneous CD-like pathology without intervention	Low breeding efficiency; variable disease onset
Microbiome-Induced (FMT) [189,190]	FMT into IL-10-/- or C57BL/6J mice	Microbiota-driven inflammation, weight loss	Allows study of host-microbe interactions	Contamination risk; strain-level microbial ID is difficult

## Data Availability

No new data were created or analyzed in this study. Data sharing does not apply to this article.

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
