# Peer review of "From AI-Assisted In Silico Computational Design to Preclinical In Vivo Models: A Multi-Platform Approach to Small Molecule Anti-IBD Drug Discovery"

_pharmaceuticals, 2025, doi:10.3390/ph18101536_

Round 1

Reviewer 1 Report

Comments and Suggestions for Authors

The manuscript submitted by the authors represents an extensive and detailed review of drug design and biological evaluation strategies for the treatment of inflammatory bowel disease (IBD). The review includes a total of 188 references, which in the opinion of this reviewer is more than appropriate to qualify as a comprehensive bibliographic analysis. Most of the cited works correspond to studies from the past decade, although a few notable older references are also included. I offer the following comments with the aim of improving and refining the presentation of the discussion.

  1. The authors employed four tables to systematize the information, which is appropriate, as it allows rapid access to specific data. However, the final table is labeled as “Table 8,” even though it is the fourth table in the manuscript. This numbering error, as well as the corresponding citations in the text, should be corrected.

  2. In my view, the number of figures is insufficient for a manuscript of this length. Figure 1 is straightforward and provides useful information about the keywords employed in the literature search. Figure 2, although visually well-prepared, contributes little to the presentation or discussion of the reviewed findings. While this image could be retained (since the authors invested effort in creating it using BioRender), I believe the manuscript would benefit substantially from the inclusion of additional figures. For example, a figure illustrating the chemical structures of the most important drugs currently used in IBD, together with information about their molecular targets or immediate clinical applications, would add significant value.

  3. In the section where molecular docking applications are discussed, there is no mention of SwissDock. Could the authors address this platform and comment on whether it has been successfully applied in the context of IBD drug discovery?

Other comments: 

Title = In scientific style, Latin phrases are lower-case and not hyphenated (“in silico”, “in vivo”).

Abstract = "critical important" --> "critical importance"

Methods = there are large numbers with no criteria. Please add explicit inclusion/exclusion criteria.

Table 2 = is Chimera software for docking calculation o just visualitazion?

Author Response

Review 1 Report Form

The manuscript submitted by the authors represents an extensive and detailed review of drug design and biological evaluation strategies for the treatment of inflammatory bowel disease (IBD). The review includes a total of 188 references, which in the opinion of this reviewer is more than appropriate to qualify as a comprehensive bibliographic analysis. Most of the cited works correspond to studies from the past decade, although a few notable older references are also included. I offer the following comments with the aim of improving and refining the presentation of the discussion.

  1. The authors employed four tables to systematize the information, which is appropriate, as it allows rapid access to specific data. However, the final table is labelled as “Table 8,” even though it is the fourth table in the manuscript. This numbering error, as well as the corresponding citations in the text, should be corrected.

Thank you.  We have corrected the table numbers and corresponding citation.

  1. In my view, the number of figures is insufficient for a manuscript of this length. Figure 1 is straightforward and provides useful information about the keywords employed in the literature search. Figure 2, although visually well-prepared, contributes little to the presentation or discussion of the reviewed findings. While this image could be retained (since the authors invested effort in creating it using BioRender), I believe the manuscript would benefit substantially from the inclusion of additional figures. For example, a figure illustrating the chemical structures of the most important drugs currently used in IBD, together with information about their molecular targets or immediate clinical applications, would add significant value.

Thank you for the comment regarding the limited number of figures. In response, an additional figure has been incorporated into the revised manuscript to strengthen the presentation of the workflow and overall structure of the review.

The suggestion to include chemical structures of currently used IBD drugs was carefully considered. However, since drug chemistry or structure–activity relationships were not discussed in any section of the manuscript, it was felt that the inclusion of such a figure would extend beyond the current scope of the review. Therefore, we respectfully did not address this suggestion but instead of introducing chemical structures, efforts were made to enhance the clarity and explanatory value of the existing figures.

  1. In the section where molecular docking applications are discussed, there is no mention of SwissDock. Could the authors address this platform and comment on whether it has been successfully applied in the context of IBD drug discovery?

The comment regarding the omission of SwissDock in the section on molecular docking applications was appreciated. In response, SwissDock has now been mentioned in the revised manuscript, along with a brief description of its relevance to IBD-related drug discovery.
Other comments: 

Title = In scientific style, Latin phrases are lower-case and not hyphenated (“in silico”, “in vivo”).

Corrected

Abstract = "critical important" --> "critical importance"

Corrected

Methods = there are large numbers with no criteria. Please add explicit inclusion/exclusion criteria.

Added with PRISMA diagram.

Table 2 = is Chimera software for docking calculation o just visualitazion?

Edited the software details addressing another reviewer comment.

Reviewer 2 Report

Comments and Suggestions for Authors

The manuscript provides a review of small-molecule discovery in IBD, covering in silico, in vitro, and in vivo strategies. The topic is timely, and the broad integration is valuable, but the review remains mostly descriptive and lacks methodological rigor in how studies were selected. Overall, the paper is promising but requires major revision for accuracy, reproducibility, and clarity.

Major suggestions:

  1. Define clearly whether this is a systematic, scoping, or narrative review, and provide transparent details of search strategy, inclusion/exclusion criteria, and screening process.
  2. Replace the current database hit counts with a PRISMA style flowchart that shows how the final set of studies was chosen.
  3. Revise the Table 2. At present, these mix visualization tools, docking engines, and simulation software, and in some cases misstate algorithms. For example, PyMOL and UCSF Chimera are visualization platforms, not docking engines. These tables should be reorganized into categories with corrected descriptions.
  4. Improve the Figure 1 to present information that is genuinely useful, for example, a schematic of the workflow or an evidence map.
  5. Add one or two case studies that demonstrate how integrated approaches have advanced IBD drug discovery.
  6. Organize the biomarker section more clearly by distinguishing diagnostic, predictive, and monitoring markers.
  7. Include a discussion of validation and reproducibility standards, noting what should be expected from computational models, organoid systems, and animal studies.
  8. Moderate the overstated claims about the impact of AI/ML and multi-omics and align language with the current level of evidence
  9. Revise the conclusions to move beyond general statements and instead highlight specific, actionable recommendations for future research.

Minor suggestions:

  1. Standardize the naming of cell lines, assays, and abbreviations, and correct factual errors such as cell line names.
  2. Keep terminology, units, and symbols consistent throughout the text.
  3. Shorten overly long sentences and reduce repetition to improve readability.
  4. Edit the figure legends so they are self-contained but concise.
  5. Review grammar, spacing, and formatting for overall polish.

Author Response

Review 2 Report Form

Comments and Suggestions for Authors

The manuscript provides a review of small-molecule discovery in IBD, covering in silico, in vitro, and in vivo strategies. The topic is timely, and the broad integration is valuable, but the review remains mostly descriptive and lacks methodological rigor in how studies were selected. Overall, the paper is promising but requires major revision for accuracy, reproducibility, and clarity.

Major suggestions:

  1. Define clearly whether this is a systematic, scoping, or narrative review, and provide transparent details of search strategy, inclusion/exclusion criteria, and screening process.

Thank you. We corrected and provided transparent detail of search strategy, inclusion and exclusion criteria and screening process.

  1. Replace the current database hit counts with a PRISMA style flowchart that shows how the final set of studies was chosen.

Added

  1. Revise the Table 2. At present, these mix visualization tools, docking engines, and simulation software, and in some cases misstate algorithms. For example, PyMOL and UCSF Chimera are visualization platforms, not docking engines. These tables should be reorganized into categories with corrected descriptions.

Thank you. The table has been fully revised and reorganized into separate categories, including docking engines, molecular visualization tools, and integrated platforms. Descriptions have been corrected accordingly to prevent any misrepresentation of algorithms or functionalities. For example, PyMOL and UCSF Chimera are now clearly identified as visualization platforms rather than docking engines.

  1. Improve the Figure 1 to present information that is genuinely useful, for example, a schematic of the workflow or an evidence map.

The recommendation to improve Figure 1 for greater informational value was acknowledged. Since other reviewer stated that Figure 1 was visually clear and well-aligned with the keyword strategy, we respectfully did not address this change. However, to address the concern raised, a more self-explanatory caption has been added to clarify its relevance. Furthermore, an additional figure has been included to provide a schematic of the workflow/evidence mapping, thereby improving overall clarity for the reader.

  1. Add one or two case studies that demonstrate how integrated approaches have advanced IBD drug discovery.

We appreciate the recommendation to include one or two case studies demonstrating the impact of integrated approaches in advancing IBD drug discovery. In response, relevant case studies have been added to the revised manuscript, supported with appropriate references, to illustrate how combined in silico, in vitro, and in vivo strategies have facilitated the development of successful therapeutic candidates.

  1. Organize the biomarker section more clearly by distinguishing diagnostic, predictive, and monitoring markers.

This section has been restructured accordingly, with each biomarker now classified under its respective category to improve clarity and readability.

  1. Include a discussion of validation and reproducibility standards, noting what should be expected from computational models, organoid systems, and animal studies.

The suggestion to address validation and reproducibility standards across computational, organoid, and animal models was acknowledged. Accordingly, a brief section has been added highlighting that candidate compounds should be screened consistently across in silico, in vitro, and in vivo systems, with feedback loops incorporated to ensure that experimental outcomes are used to refine computational predictions. This iterative cross-validation has been emphasized as essential for reproducibility and translational reliability.

  1. Moderate the overstated claims about the impact of AI/ML and multi-omics and align language with the current level of evidence

Addressed the comment in the respective sections.

  1. Revise the conclusions to move beyond general statements and instead highlight specific, actionable recommendations for future research.

The suggestion to revise the conclusion section to move beyond general statements and provide more specific, actionable recommendations for future research was appreciated. In response, the conclusion has been rewritten to emphasize clear research priorities and practical directions for advancing IBD drug discovery through integrated computational, experimental, and translational approaches.

Minor suggestions:

  1. Standardize the naming of cell lines, assays, and abbreviations, and correct factual errors such as cell line names.

Corrected.

  1. Keep terminology, units, and symbols consistent throughout the text.

Edited.

  1. Shorten overly long sentences and reduce repetition to improve readability.

Edited.

  1. Edit the figure legends so they are self-contained but concise.

Edited.

  1. Review grammar, spacing, and formatting for overall polish.

Edited.

Reviewer 3 Report

Comments and Suggestions for Authors

Review of the Manuscript

Title: From AI-assisted In-Silico Computational Design to Preclinical In-Vivo Models: A Multi-platform Approach to Small Molecule Anti-IBD Drug Discovery

The review successfully covers the continuum of IBD drug discovery: from computational methods (docking, QSAR, network pharmacology, AI) to in vitro models (cell lines, organoids, OoC) and in vivo approaches.

However, there are some points to improve:

  • Include a schematic “pipeline” figure showing how in silico → in vitro → in vivo models integrate, with feedback loops. This would make the review more didactic.
  • Instead of only listing software/databases, provide a critical appraisal (e.g., which docking tools are most reproducible, which organoid systems best model IBD heterogeneity).
  • Add discussion of how AI-driven or organoid-based drug discovery has already informed early-phase clinical trials in IBD.
  • A dedicated section highlighting gaps and opportunities (e.g., patient stratification, integration of microbiome–host modelling, ethical issues in AI) would strengthen the conclusion.
  • While the manuscript is well-referenced, ensure a balance of landmark papers and the most recent studies. Some areas could benefit from more critical referencing (e.g., limitations of docking in flexible proteins).

Author Response

Review 3 Report Form

Comments and Suggestions for Authors

Review of the Manuscript

Title: From AI-assisted In-Silico Computational Design to Preclinical In-Vivo Models: A Multi-platform Approach to Small Molecule Anti-IBD Drug Discovery

The review successfully covers the continuum of IBD drug discovery: from computational methods (docking, QSAR, network pharmacology, AI) to in vitro models (cell lines, organoids, OoC) and in vivo approaches.

However, there are some points to improve:

  • Include a schematic “pipeline” figure showing how in silico → in vitro → in vivo models integrate, with feedback loops. This would make the review more didactic.

Thank you. We have added and corrected  Figures 3 and 4.

  • Instead of only listing software/databases, provide a critical appraisal (e.g., which docking tools are most reproducible, which organoid systems best model IBD heterogeneity).

The suggestion to include a schematic pipeline demonstrating the integration of in silico → in vitro → in vivo models with feedback loops was appreciated. In response, such a schematic has been incorporated (Figure 4) into the revised manuscript to enhance the didactic value of the review. The newly added figure visually illustrates the translational continuum between computational screening, experimental validation, and preclinical assessment, thereby reinforcing the conceptual framework discussed throughout the manuscript.

  • Add discussion of how AI-driven or organoid-based drug discovery has already informed early-phase clinical trials in IBD.

Addressed in 3.2.7 and  4.2.2 sections

  • A dedicated section highlighting gaps and opportunities (e.g., patient stratification, integration of microbiome–host modelling, ethical issues in AI) would strengthen the conclusion.

A section addressing reviewer comments, highlighting gaps and opportunities such as patient stratification, microbiome–host integration, and AI ethics has been added. This section has been addressed by merging with other reviewer comment.

  • While the manuscript is well-referenced, ensure a balance of landmark papers and the most recent studies. Some areas could benefit from more critical referencing (e.g., limitations of docking in flexible proteins).

Added and addressed the limitations of docking in flexible proteins

Reviewer 4 Report

Comments and Suggestions for Authors

A major revision of the manuscript is necessary before the manuscript is recommended for publication in Pharmaceuticals.

In this review, the authors summarize recent strategies for drug discovery in inflammatory bowel disease (IBD), which includes ulcerative colitis and Crohn’s disease. They conducted an extensive literature search covering 188 studies from 2000-2025 and integrated findings across computational modelling, molecular docking, network pharmacology, and bioinformatics approaches. The work highlights how in silico methods can accelerate target identification and screening, while also stressing the complementary use of 2D/3D cell cultures and advanced animal models to validate translational potential. The review further emphasizes the need for linking computational predictions with experimental evidence to improve the reliability of therapeutic development. Overall, this paper presents a multi-platform paradigm that bridges computer modelling, cell-based studies, and in vivo validation, aiming to improve drug discovery for IBD and to support more personalized therapeutic approaches. Several aspects, however, require further clarification and refinement to enhance its scientific rigor and practical relevance.

  1. In the Review Methodology section, the manuscript outlines the search strategy and keywords but does not specify the inclusion and exclusion criteria for selecting studies. Clarifying criteria such as study design, sample size, language restrictions, and quality assessment, as well as reporting the final number of included studies, would improve transparency and reproducibility.
  2. In the 3.1 Discovery of Novel Biomarkers in IBD section, the manuscript lists various omics-based biomarkers and refers to Table 1 but does not explain the selection criteria. Indicating whether biomarkers were chosen based on clinical validation, literature prevalence, or novelty, and distinguishing those already applied clinically from experimental markers, would enhance clarity and applicability.
  3. In Table 3, the website link in row 7 is bolded inconsistently; formatting should be standardized.
  4. In the Conclusion and Future Direction section, the manuscript emphasizes integrating in silico predictions with advanced in vitro and in vivo models but lacks specific guidance on future research priorities. Expanding this section with concrete experimental strategies, such as prioritizing patient-derived models or single-cell applications, would provide a clearer roadmap for advancing IBD drug discovery.
  5. Ensure consistent capitalization in reference titles (e.g., refs. 9 & 34) and refine English for precision, fluency, and unambiguous expression of aims, methods, results, and conclusions.
Comments on the Quality of English Language

The English could be improved to more clearly express the research.

Author Response

Review 4 Report Form

Open Review

Comments and Suggestions for Authors

A major revision of the manuscript is necessary before the manuscript is recommended for publication in Pharmaceuticals.

In this review, the authors summarize recent strategies for drug discovery in inflammatory bowel disease (IBD), which includes ulcerative colitis and Crohn’s disease. They conducted an extensive literature search covering 188 studies from 2000-2025 and integrated findings across computational modelling, molecular docking, network pharmacology, and bioinformatics approaches. The work highlights how in silico methods can accelerate target identification and screening, while also stressing the complementary use of 2D/3D cell cultures and advanced animal models to validate translational potential. The review further emphasizes the need for linking computational predictions with experimental evidence to improve the reliability of therapeutic development. Overall, this paper presents a multi-platform paradigm that bridges computer modelling, cell-based studies, and in vivo validation, aiming to improve drug discovery for IBD and to support more personalized therapeutic approaches. Several aspects, however, require further clarification and refinement to enhance its scientific rigor and practical relevance.

  1. In the Review Methodology section, the manuscript outlines the search strategy and keywords but does not specify the inclusion and exclusion criteria for selecting studies. Clarifying criteria such as study design, sample size, language restrictions, and quality assessment, as well as reporting the final number of included studies, would improve transparency and reproducibility.

Thank you. We have added a PRISMA style diagram.

  1. In the 3.1 Discovery of Novel Biomarkers in IBD section, the manuscript lists various omics-based biomarkers and refers to Table 1 but does not explain the selection criteria. Indicating whether biomarkers were chosen based on clinical validation, literature prevalence, or novelty, and distinguishing those already applied clinically from experimental markers, would enhance clarity and applicability.

Thank you. The biomarkers summarized in Table 1 were selected based on a comprehensive literature search, focusing on current clinical and translational research, as well as potential future applications. Selection criteria considered clinical validation, prevalence in the literature, and novelty, ensuring a clear distinction between biomarkers already applied in clinical practice, those under early clinical validation, and emerging experimental markers.

  1. In Table 3, the website link in row 7 is bolded inconsistently; formatting should be standardized.

Corrected

  1. In the Conclusion and Future Direction section, the manuscript emphasizes integrating in silico predictions with advanced in vitro and in vivo models but lacks specific guidance on future research priorities. Expanding this section with concrete experimental strategies, such as prioritizing patient-derived models or single-cell applications, would provide a clearer roadmap for advancing IBD drug discovery.

Addressed in the last section of the conclusion.

  1. Ensure consistent capitalization in reference titles (e.g., refs. 9 & 34) and refine English for precision, fluency, and unambiguous expression of aims, methods, results, and conclusions.

Corrected and edited all the references, all the titles are now in similar patterns. Refined English accordingly.

Round 2

Reviewer 1 Report

Comments and Suggestions for Authors

The authors have addressed the comments and now the manuscript is suitable for publication.

Reviewer 2 Report

Comments and Suggestions for Authors

The authors have addressed the queries, and their study can now be considered for acceptance.

Reviewer 4 Report

Comments and Suggestions for Authors

This manuscript is recommended for acceptance.